# On the Hardness of Approximating Distributions with Probabilistic Circuits

**John Leland**[1]          **YooJung Choi**[1]

[1]School of Computing and Augmented Intelligence, Arizona State University, Tempe, Arizona, USA

## Abstract

A fundamental challenge in probabilistic modeling is balancing expressivity and tractable inference. Probabilistic circuits (PCs) aim to directly address this tradeoff by imposing structural constraints that guarantee efficient inference of certain queries while maintaining expressivity. Since inference complexity on PCs depends on circuit size, understanding the size bounds across circuit families is key to characterizing the tradeoff between tractability and expressive efficiency. However, expressive efficiency is often studied through exact representations, where exactly encoding distributions while enforcing various structural properties often incurs exponential size blow-ups. Thus, we pose the following question: *can we avoid such size blow-ups by allowing some small approximation error?* We first show that approximating an arbitrary distribution with bounded $f$-divergence is NP-hard for any model that can tractably compute marginals. We then prove an exponential size gap for approximation between the class of decomposable PCs and additionally deterministic PCs.

## 1 INTRODUCTION

Generative models have shown remarkable success in capturing complex distributions [21, 24, 29, 22, 39]; yet, despite their expressivity, they often do not support efficient computation of fundamental probabilistic queries, which are critical for inference in domains such as healthcare [34], neuro-symbolic AI [37], algorithmic fairness [7], and environmental science [3]. Probabilistic circuits address this by balancing expressivity and tractable inference. Naturally, there have been many works characterizing the expressive efficiency of different circuit classes [16, 1, 6, 42, 43, 17]. However, how structural constraints affect the ability to *approximate* distributions remains underexplored.

This paper studies the following fundamental question: *does allowing small approximation error alleviate the exponential separation between circuit classes observed in exact modeling, or does hardness persist even in the approximate setting?* Our motivation is two-fold. (1) Showing that certain distributions cannot be approximated within a bounded distance compactly by a family of PCs implies that any learning algorithm whose hypothesis space is that family of PCs would fail to learn the distribution with a bounded approximation error. (2) Moreover, PCs can also be used to perform inference on other probabilistic models (such as Bayesian networks or probabilistic programs) by compiling them into PCs then performing inference on the compiled circuits [2, 23, 14, 5, 20]. This suggests the following approximate inference scheme: *approximately* compile a probabilistic model into a PC then run efficient *exact* inference on the approximately compiled PC. Thus, if we can bound the distance between the target distribution and approximate model, we can hope to provide guarantees on the approximate inference results as well.

Our main contributions are as follows: (1) we prove that it is NP-hard to approximate distributions within a bounded $f$-divergence using any model that tractably computes marginals, via a reduction from SAT (Theorem 3.3 and 3.4); (2) we derive an unconditional, exponential separation between decomposable PCs and decomposable & deterministic PCs for approximate modeling (Theorem 4.1); (3) we study the relationship between bounds on divergence measures for approximate modeling and approximation errors for marginal and maximum-a-posteriori (MAP) inference, characterizing when one is or is not sufficient to guarantee the other (Section 3 and further discussion in Appendix B).

## 2 PRELIMINARIES

**Notations**    We use uppercase letters ($X$) to denote random variables and lowercase letters ($x$) to denote their assignments. Sets of random variables and assignments are denoted using bold letters ($\mathbf{X}$ and $\mathbf{x}$). The *accepting models* of

*Accepted for the $8^{th}$ Workshop on Tractable Probabilistic Modeling at UAI  (TPM 2025).*

a Boolean function $f : \{0,1\}^n \to \{0,1\}$ over $n$ variables is denoted by $f^{-1}(1)$, and the number of accepting models by $\mathrm{MC}(f)$. Moreover, a Boolean function $f$ is the *support* of a distribution $P$, if $P$ is positive only over the models of $f$.

**Probabilistic Circuits** A *probabilistic circuit (PC)* recursively encodes a joint probability distribution over random variables through a parameterized directed acyclic graph (DAG), composed of 3 types of nodes: leaf, product $\otimes$, and sum nodes $\oplus$. PCs are closely related to *logical circuits* in the *knowledge compilation* literature [16]. Logical circuits encode Boolean functions as directed acyclic graphs (DAGs) consisting of AND ($\wedge$) and OR ($\vee$) gates—analogous to $\otimes$ and $\oplus$ nodes, respectively—with positive and negative literals as leaf nodes. We can characterize different families of both probabilistic and logical circuits based on their structural properties, such as *decomposability* and *determinism*. These circuit properties allow tractable (polytime) computation of various queries); for instance, decomposable PCs admit linear-time *marginal* inference while deterministic and decomposable PCs additionally allow linear-time *maximum-a-posteriori (MAP)* inference. There is a rich literature studying different families of circuit representations in terms of their tractability for inference and operations, as well as their relative *succinctness (expressive efficiency)* for both exact and approximate compilation, which we will leverage for our hardness results and size lower bounds on PCs. See Appendix A for further background on PCs.

**Measures of Difference between Probability Distributions** To study the hardness of approximating probability distributions, we must first measure how "good" an approximation is. We focus on the class of $f$-divergences.

**Definition 2.1** ($f$-divergence [30]). Let $f : (0, \infty) \to \mathbb{R}$ be a convex function with $f(1) = 0$, and $P, Q$ be two probability distributions over a set of Boolean variables $\mathbf{X}$. If $Q > 0$ wherever $P > 0$, the *$f$-divergence* between $P$ and $Q$ is defined as $D_f(P\|Q) = \sum_{\mathbf{x}} Q(\mathbf{x}) f(\frac{P(\mathbf{x})}{Q(\mathbf{x})})$.

Commonly used $f$-divergences include the Kullback-Leibler divergence, $\chi^2$-divergence, and the total variation distance, which is especially relevant to our results.

**Definition 2.2** (Total Variation Distance). The *total variation distance (TVD)* between two probability distributions $P$ and $Q$ over a set of $n$ Boolean variables $\mathbf{X}$ is defined as $D_{\mathsf{TV}}(P\|Q) = \frac{1}{2} \sum_{\mathbf{x} \in \mathbf{X}} |P(\mathbf{x}) - Q(\mathbf{x})|$, or equivalently $D_{\mathsf{TV}}(P\|Q) = \max_{S \subseteq \{0,1\}^n} |P(S) - Q(S)|$.

To more compactly describe PCs that approximate distributions under some bounded distance, we use the following.

**Definition 2.3** ($\epsilon$-$D$-Approximation). Let $P, Q$ be some probability measures and $D$ be some distance we want to minimize between $P$ and $Q$. We say that $Q$ is an $\epsilon$-$D$-Approximator of $P$ if $D(P\|Q) < \epsilon$ for some $\epsilon > 0$.

We refer to any circuit $Q$ such that it approximates our target distribution $P$ within $D_{\mathsf{TV}}(P\|Q) < \epsilon$ a $\epsilon$-$\mathcal{D}_{\mathsf{TV}}$-*Approximator*. The majority of our results are derived using properties of the total variation distance, due to its properties as a distance metric. To extend our results to $f$-divergences, we utilize the class of $k$-convex f-divergences, which provides an upper bound on the total variation distance.

We can define a $k$-convex f-divergence [28] as a $\mathbb{R} \cup \{\infty\}$-valued function $f$ on a convex set $K \subseteq \mathbb{R}$ is *k-convex* when $x, y \in K$ and $t \in [0, 1]$ implies $f((1 - t)x + ty) \leq (1 - t)f(x) + tf(y) - kt(1 - t)\frac{(x-y)^2}{2}$. When $f$ is twice differentiable, this is equivalent to $f''(x) \geq k$ for $x \in K$. In the case that $k = 0$ this reduces to the normal notion of convexity. An $f$-divergence $D_f$ is *k-convex* over an interval $K$ for $k \geq 0$ when the function $f$ is $k$-convex on $K$.

We provide a table in Appendix C summarizing which $f$-divergence measures are $k$-convex and for which value of $k$. Throughout this paper, we express approximation bounds using $k$-convex $f$-divergences as they naturally encapsulate bounds on many common distance measures. For instance, for any $k$-convex $f$-divergence between $P$ and $Q$, we have that $D_{\mathsf{TV}}(P\|Q)^2 < D_f(P\|Q)/k$ [28]. For KL-divergence, which is the most commonly used objective for learning probabilistic models, we can use a form of Pinsker's inequality [38] to obtain $D_{\mathsf{TV}}(P\|Q) < \sqrt{\frac{1}{2}D_{\mathsf{KL}}(P\|Q)}$.

## 3 APPROXIMATE MODELING WITH TRACTABLE MARGINALS IS NP-HARD

Most works characterizing the expressive efficiency of different circuit classes have been concerned with *exact* representations [16, 1, 6, 42, 43, 17]. While Chubarian and Turán [10] and De Colnet and Mengel [18] have recently studied the ability (and hardness) of logical circuit classes to compactly *approximate* Boolean functions, to the best of our knowledge, our results are the first to show hardness of *compactly approximating probability distributions* for different families of tractable PCs.

As noted earlier, the complexity of approximately modeling distributions with PCs is valuable for understanding: (1) potential limitations in the hypothesis space of PC learning algorithms, and (2) the feasibility of approximate inference with guarantees through approximate compilation. This section aims to answer this, focusing on probabilistic models that are tractable for marginal queries. We first show that a form of approximate marginal inference using this scheme requires a non-trivial bound on the total variation distance between the target distribution and the approximate model, and prove that finding such an approximator is NP-hard.

We consider *relative (multiplicative) approximation* of marginal queries. Let $P(\mathbf{X})$ be a probability distribution over a set of variables $\mathbf{X}$. Then we say another distribution $Q(\mathbf{X})$ is a *relative approximator* of marginals of $P$ w.r.t.

$0 \leq \epsilon \leq 1$ if: $\frac{1}{1+\epsilon} \leq \frac{P(\mathbf{y})}{Q(\mathbf{y})} \leq 1 + \epsilon$ for every assignment $\mathbf{y}$ to a subset $Y \subseteq \mathbf{X}$. Relative approximation of marginals is known to be NP-hard for Bayesian networks [12], thus it immediately follows that approximately representing arbitrary distributions using polynomial-sized PCs tractable for marginals such that the PC is a relative approximator of all marginals is also NP-hard. However, approximating *all* marginal queries is quite a strong condition, and we may still want to closely approximate distributions as they could be useful in approximating *some* marginal queries.

We show that relative approximation for all marginal queries implies a non-trivially bounded total variation distance.

**Theorem 3.1** (Relative Approximation implies Bounded $D_{\mathsf{TV}}(P\|Q)$). *Let $\epsilon > 0$ and $P, Q$ be two probability distributions over $\mathbf{X}$. If $Q$ is a relative approximator of marginals for $P$, then $D_{\mathsf{TV}}(P\|Q) \leq \frac{\epsilon}{2}$.*

As $Q$ is a relative approximator of $P$, for all assignment $\mathbf{x}$ we have $\frac{1}{1+\epsilon} \leq \frac{P(\mathbf{x})}{Q(\mathbf{x})} \leq 1 + \epsilon$ which implies $|P(\mathbf{x}) - Q(\mathbf{x})| \leq \epsilon \min(P(\mathbf{x}), Q(\mathbf{x}))$. Therefore, $D_{\mathsf{TV}}(P\|Q) = \frac{1}{2}\sum_{\mathbf{x}} |P(\mathbf{x}) - Q(\mathbf{x})| \leq \frac{1}{2}\sum_{\mathbf{x}} \epsilon \min(P(\mathbf{x}), Q(\mathbf{x})) \leq \frac{\epsilon}{2}$.

In other words, $D_{\mathsf{TV}}(P\|Q) \leq \epsilon/2$ is a necessary condition for $Q$ to be a relative approximator of marginals of $P$ w.r.t. $\epsilon$. However, this is not a sufficient condition as shown below.

**Proposition 3.2** (Bounded $D_{\mathsf{TV}}(P\|Q)$ does not imply relative approximation). *There exists a family of distributions $P$ that have $\epsilon$-$D_{\mathsf{TV}}$-approximators, yet for any such approximator $Q$, the relative approximation error of marginals between $P$ and $Q$ can be arbitrarily large.*

We prove the above proposition by explicitly constructing a family of distributions $\mathcal{Q}$ such that every $Q \in \mathcal{Q}$ is an $\epsilon$-$D_f$-approximator for any arbitrary $\epsilon > 0$ and distribution $P$ yet $P(\mathbf{x})/Q(\mathbf{x})$ can be arbitrarily large for some $\mathbf{x}$. See Appendix E.1 for the full construction.

Since approximating a distribution $P$ with a bounded TV distance is a necessary but insufficient condition for the NP-hard problem of relative approximation of marginals, this raises the question whether it is still possible to efficiently approximate the distribution $P$ with a compact PC $Q$ that is tractable for marginals. We next answer this in the negative.

**Theorem 3.3** (Hardness of $D_f$-approximation). *Given a probability distribution $P$ and a $k$-convex $f$-divergence $D_f$, for any $0 < \epsilon < \frac{1}{4}$, it is NP-hard to represent its $k\epsilon^2$-$D_f$-approximation as a model that admits tractable marginals.*

For the full proof, see Appendix E.2. We prove this using a reduction to SAT. First, we construct a formula $f$ over $\mathbf{X}$, and another formula $f'$ over $\mathbf{X}$ and $Y$ where $f' = (Y \wedge f) \vee (\neg Y \wedge X_1 \wedge \cdots \wedge X_n)$. Let $P$ be uniform over the models of $f'$ and $Q$ be a model such that $D_f(P\|Q) < k\epsilon^2$ with $0 \leq$

$\epsilon < \frac{1}{4}$. Then, using bounded difference between marginals, we show that $f$ is satisfiable if and only if $Q(Y = 1) \geq \frac{1}{4}$.

**Corollary 3.4.** *Given a probability distribution $P$, for $0 < \epsilon < \frac{1}{4}$, it is NP-hard to represent its $\epsilon$-$D_{\mathsf{TV}}$-approximation as a model that can tractably compute marginals.*

Thus, the class of polynomial-sized models which are tractable for marginals [44, 26, 40, 6], cannot contain $k\epsilon^2$-$D_f$-approximations for all distributions unless $\mathsf{P} = \mathsf{NP}$.

While Martens and Medabalimi [27] previously showed a related result that there exists a function for which a sequence of decomposable PCs converging to approximate the function arbitrarily well requires an exponential size, our result applies more broadly to any class of models supporting tractable marginals as well as allowing for bounded but non-vanishing approximation error. Thus, representing functions or distributions as decomposable PCs, even approximately, remains hard, challenging the feasibility of approximate compilation for approximate inference with small error.

## 4 LARGE DEC. & DET. PCS FOR APPROXIMATE MODELING

Continuing our characterization of approximation power of PCs, we now turn to the family of decomposable and deterministic PCs. We take inspiration from related results for logical circuits. In particular, Bova et al. [1] proved an exponential separation of DNNFs from d-DNNFs;[1] De Colnet and Mengel [18] showed that there exists functions which require exponentially sized d-DNNF to approximate.

Nevertheless, this does not immediately imply the same separation for probabilistic circuits for two main reasons: (1) approximation for PCs is measured in terms of divergences between distributions rather than some probabilistic error between Boolean functions, and (2) our approximator can represent arbitrary distributions instead of being limited to a Boolean function (or a uniform distribution over it). This section presents our proof of exponential separation of decomposable PCs from decomposable & deterministic ones, by constructing a family of distributions that can be represented by compact decomposable PCs but any PC that is also deterministic and approximates it within a bounded TV distance requires exponential size.

We consider the *Sauerhoff function* [35] which was used to show the separation between DNNFs and d-DNNFs for exact compilation [1]. Let $g_n : \{0,1\}^n \to \{0,1\}$ be a function evaluating to 1 if and only if the sum of its inputs is divisible by 3. The *Sauerhoff function* is defined as $S_n : \{0,1\}^{n^2} \to \{0,1\}$ over the $n \times n$ matrix $X = (x_{ij})_{1 \leq i,j \leq n}$ by $S_n(X) = R_n(X) \vee C_n(X)$ where $R_n, C_n : \{0,1\}^{n^2} \to \{0,1\}$ are

---

[1]Decomposable negation normal forms (DNNFs) and deterministic, decomposable negation normal forms (d-DNNFs).

defined by $R_n(X) = \bigoplus_{i=1}^{n} g_n(x_{i,1}, x_{i,2}, \ldots, x_{i,n})$ and $C_n(X) = R_n(X^T)$. Here, $\oplus$ represents addition modulo 2.

There exists a DNNF of size $O(n^2)$ that exactly represents the Sauerhoff function $S_n$, constructed using two compact ordered binary decision diagrams to represent $R_n, C_n$ [1, Proposition 7]. We then define our family of target distributions $P_n$ as follows: let $\mathcal{C}_n$ be a DNNF for $S_n$ with size $O(n^2)$; then $P_n$ is a decomposable PC obtained by replacing the literals of $\mathcal{C}_n$ with corresponding indicator functions, $\vee$ with $\oplus$, and $\wedge$ with $\otimes$ nodes with uniform parameters, then smoothing. Note that $P_n(\mathbf{x}) > 0$ if and only if $S_n(\mathbf{x}) = 1$. We will show that a deterministic, decomposable PC approximating $P_n$ requires exponential size.

**Theorem 4.1** (Exponential-Size Deterministic PC). *A deterministic, decomposable PC that is a $\epsilon$-$D_{\mathsf{TV}}$-Approximator of $P_n$, where $\epsilon = \frac{1}{16} - \Omega(1/Poly(n^2))$, has size $2^{\Omega(n)}$.*

We will prove the above by first showing that approximation of $P_n$ with a deterministic and decomposable PC implies a form of *weak approximation* [18] of $S_n$ with a d-DNNF of the same size, and next proving that such d-DNNF must be exponentially large.

**Definition 4.2** (Weak Approximation [18]). We say that a Boolean formula $g$ is a weak $\epsilon$-approximation of another Boolean formula $f$ if $\mathrm{MC}(f \wedge \neg g) + \mathrm{MC}(\neg f \wedge g) \le \epsilon 2^n$.

**Proposition 4.3** (Bounded $D_{\mathsf{TV}}$ implies weak approximation). *Let $0 \le \epsilon < \frac{1}{8}$ and $P$ be a uniform distribution whose support is given by a Boolean function $f$. Suppose that $Q$ is a decomposable and deterministic PC representing an $\epsilon$-$D_{\mathsf{TV}}$-Approximator of $P$. Then there exists a d-DNNF $g$ which has size polynomial in the size of $Q$ that represents a $4\epsilon$-weak-approximator of $f$.*

For full proof see Appendix E.4. Start with our circuit $Q$ with support $g$, and use Algorithm E.3 to prune $Q(\mathbf{x}) < \frac{1}{2^{n+1}}$. We call this unnormalized distribution $Q'$ and it's support $g'$. Using $D_{\mathsf{TV}}(P \| Q) < \epsilon$, the fact that on $g'$ all $Q(\mathbf{x}) \ge \frac{1}{2^{n+1}}$, and that $\mathrm{MC}(f) \le 2^n$, we show that we can then bound $\mathrm{MC}(f \wedge \neg g') + \mathrm{MC}(\neg f \wedge g') < 4\epsilon 2^n$, which corresponds to a $4\epsilon$-weak-approximation.

**Proposition 4.4** (d-DNNF approximating $S_n$ has exponential size). *A d-DNNF representing a $(\frac{1}{4} - \Omega(1/\mathrm{Poly}(n^2)))$-weak-approximation of $S_n$ has size $2^{\Omega(n)}$.*

*Proof.* Let $\mathcal{C}$ be a d-DNNF that is a $(\frac{1}{4} - \Omega(1/\mathrm{Poly}(n^2)))$-weak-approximation of $S_n$. Sauerhoff [35] showed that any "two-sided" rectangle approximation[2] (matching the notion of weak approximation) of $S_n$ within $\frac{1}{4} - \Omega(1/\mathrm{Poly}(n^2))$ must have size $2^{\Omega(n)}$. Bova et al. [1] further showed that a d-DNNF $\mathcal{C}$ computing a function $f$ is a balanced rectangle partition of $f$ with size at most $|\mathcal{C}|$. Thus, $|\mathcal{C}| = 2^{\Omega(n)}$. $\square$

---

[2]See Appendix D for details on rectangle partitions

Since by we can represent $S_n$ by a DNNF of size $O(n^2)$, there exists an exponential gap in approximation between DNNF and d-DNNF. We next show this exponential gap in approximation persists for PCs.

*Proof Sketch of Theorem 4.1.* See Appendix E.5 for the full proof. Suppose we have a deterministic, decomposable PC $Q$ that is an $\epsilon$-$D_{\mathsf{TV}}$-Approximator of $P_n$ for $\epsilon = (\frac{1}{16} - \Omega(1/Poly(n^2)))$. Consider the uniform distribution $U$ over $S_n$. We use the triangle inequality to bound $D_{\mathsf{TV}}(U \| Q) \le D_{\mathsf{TV}}(U \| P_n) + D_{\mathsf{TV}}(P_n \| Q)$. Given that $D_{\mathsf{TV}}(P_n \| Q) < \frac{1}{16} - \Omega(1/Poly(n^2))$, it is only left to bound $D_{\mathsf{TV}}(U \| P_n)$. Taking the d-DNNF constructed by [1, Prop. 7] and the low 0-density property of the Sauerhoff function [35], we bound $D_{\mathsf{TV}}(P_n \| U) < \eta$ for $\eta = \frac{1}{(1-1/\sqrt{2})2^{n^2}}$, which implies $D_{\mathsf{TV}}(U \| Q) < \epsilon - \Omega(1/Poly(n^2)) + \eta$. We combine $\Omega(1/Poly(n^2)) + \eta$ to simplify this expression. By Proposition 4.3, we can construct a d-DNNF $\mathcal{C}'$ from $Q$—replacing $\oplus$ and $\otimes$ with $\vee$ and $\wedge$ respectively—that is a $(\frac{1}{4} - \Omega(1/Poly(n^2)))$-weak-approximation of $S_n$. By Proposition 4.4, $|Q| = |\mathcal{C}'| = 2^{\Omega(n)}$. $\square$

As we have constructed a decomposable PC $P_n$ that has size $O(n^2)$ such that any deterministic, decomposable PC approximating it has size $O(2^{\Omega(n)})$, we have shown an unconditional exponential gap for approximation between decomposable PCs and deterministic, decomposable PCs. Thus, approximate modeling does not grant the ability to overcome exponential expressive efficiency gaps.

## 5 CONCLUSION

We established the hardness of approximating distributions with probabilistic circuits such that the $f$-divergence is small. First, we showed that this task is NP-hard for any model that is tractable for marginal inference, including decomposable PCs. Then, we used the Sauerhoff function to demonstrate an exponential gap between decomposable PCs and deterministic, decomposable PCs when allowing for bounded approximation error. This proves that the succinctness gap in the case of exact compilation persists even under relaxed approximation conditions.

These results highlight key challenges in learning compact and expressive PCs while maintaining tractable inference. In light of this, we ask: can a polynomial-time algorithm enable learning an $\epsilon$-approximator for a broad family of distributions with a more relaxed $\epsilon$? Furthermore, does there exist modeling conditions that are sufficient for relative approximation of various queries? Lastly, we see this work as a first step to encourage further theoretical studies on approximate modeling and inference with guarantees using tractable models. In this paper, we focused on PCs that are tractable for marginal and MAP queries, but there are large classes of tractable models whose efficient approximation power remain unknown.

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

# On the Hardness of Approximating Distributions with Probabilistic Circuits (Supplementary Material)

**John Leland**[1]

**YooJung Choi**[1]

[1]School of Computing and Augmented Intelligence, Arizona State University, Tempe, Arizona, USA

## A  BACKGROUND

### A.1  PROBABILISTIC CIRCUITS

Probabilistic circuits (PCs) [6] provide a unifying framework for a wide class of tractable probabilistic models, including arithmetic circuits [13], sum-product Networks [31], cutset networks [32], probabilistic sentential decision diagrams [25], and bounded-treewidth graphical models [19, 9].

**Definition A.1** (Probabilistic circuits). A probabilistic circuit (PC) $\mathcal{C} := (\mathcal{G}, \theta)$ represents a joint probability distribution $p(\mathbf{X})$ over random variables $\mathbf{X}$ through a directed acyclic graph (DAG) $\mathcal{G}$ parameterized by $\theta$. The DAG is composed of 3 types of nodes: leaf, product $\otimes$, and sum nodes $\oplus$. Every leaf node in $\mathcal{G}$ is an input, Boolean indicator functions in this paper, and every internal node receives inputs from its children $\text{in}(n)$. The scope of a given node, $\phi(n)$, is the set of random variables as inputs to $n$.

Each node $n$ of a PC is recursively defined as:

$$p_n(\mathbf{x}) := \begin{cases} l(x), & \text{if } n \text{ is a leaf} \\ \prod_{c \in \text{in}(n)} p_c(\mathbf{x}) & \text{if } n \text{ is a } \otimes \\ \sum_{c \in \text{in}(n)} \theta_{n,c} p_c(\mathbf{x}) & \text{if } n \text{ is a } \oplus \end{cases} \tag{1}$$

where $\theta_{n,c} \in [0,1]$ is the parameter associated with the edge connecting nodes $n, c$ in $\mathcal{G}$, and $\sum_{c \in \text{in}(n)} \theta_{n,c} = 1$. The distribution represented by the circuit is the output at its root node.

A key characteristic of PCs is that imposing certain structural properties on the circuit enables tractable (polytime) computation of various queries. In this paper we focus on two families of PCs: those that are tractable for *marginal* inference and for *maximum-a-posteriori (MAP)* inference. The class of marginal queries of a joint distribution $p(\mathbf{X})$ over variables $\mathbf{X}$ refers the set of functions that can compute $p(\mathbf{y})$ for some assignment $\mathbf{y}$ for $\mathbf{Y} \subseteq \mathbf{X}$. Marginalization is a fundamental statistical operation which enables reasoning about subsets of variables, essential for tasks like decision making, learning, and predicting under uncertainty. While marginal inference is #P-hard in general [33, 11], the family of PCs satisfying certain structural conditions, admit tractable marginal inference—specifically in linear time in the size of the circuit [15].

**Definition A.2** (Smoothness and Decomposability). A sum unit is *smooth* if its children have identical scopes: $\phi(c) = \phi(n), \forall c \in \text{in}(n)$. A product unit is *decomposable* if its children have disjoint scopes: $\phi(c_i) \cap \phi(c_j) = \emptyset, \forall c_i, c_j \in \text{in}(n), i \neq j$. A PC is smooth and decomposable iff every sum unit is smooth and every product unit is decomposable.

In addition, we are often interested in finding the most likely assignments given some observations. The class of maximum-

$$\theta_1 \theta_2 X_3 \neg X_3 X_2 \neg X_2 X_2 \neg X_2 X_1 \neg X_1 X_1 \neg X_1$$

Figure 1: A smooth, decomposable, and deterministic PC (weights shown only for the root for conciseness).

*Accepted for the 8th Workshop on Tractable Probabilistic Modeling at UAI* (TPM 2025).

a-posteriori (MAP)[1] queries of a joint distribution $p(\mathbf{X})$ is the set of queries that compute $\max_{\mathbf{q} \in val(\mathbf{Q})} p(\mathbf{q}, \mathbf{e})$ where $\mathbf{e} \in val(\mathbf{E})$ is an assignment to some subset $\mathbf{E} \subseteq \mathbf{X}$ and $\mathbf{Q} = \mathbf{X} \setminus \mathbf{E}$. Again, MAP inference is NP-hard in general [36] but can be performed tractably for a certain class of PCs. In particular, smoothness and decomposability are no longer sufficient, and we must enforce an additional condition.

**Definition A.3** (Determinism). A sum node is *deterministic* if, for any fully-instantiated input, the output of at most one of its children is nonzero. A PC is deterministic iff all of its sum nodes are deterministic.

Figure 1 depicts an example PC that is smooth, decomposable, and deterministic, which thus supports tractable marginal as well as MAP inference.[2]

# B    RELATIONSHIP BETWEEN APPROXIMATE MODELING AND INFERENCE

We have shown that even if we allow some approximation error, it is hard to efficiently approximate distributions using tractable probabilistic circuits. As approximating modeling is a hard task, one would hope that ensuring approximate modeling would guarantee the approximation of hard inference queries with bounded error. Unfortunately, this is not the case for all notions of approximation. Therefore, we study the relationship between approximate modeling and inference, in particular focusing on *absolute* approximation of *marginal*, *conditional*, and *maximum-a-posteriori (MAP)* queries.

In Section 3, we showed that bounded total variation distance is a necessary but insufficient condition for relative approximation of marginals. We now consider a slightly weaker notion of approximation called *absolute approximation*. Let $P(\mathbf{X})$ be a probability distribution over a set of variables $\mathbf{X}$. Then we say another distribution $Q(\mathbf{X})$ is an *absolute approximator* of marginals of $P$ w.r.t. $0 \le \epsilon \le 1$ if: $|P(\mathbf{y}) - Q(\mathbf{y})| \le \epsilon$ for every assignment $\mathbf{y}$ to a subset of variables $Y \subseteq \mathbf{X}$. We show that any model that is a $k\epsilon^2$-$D_f(P\|Q)$-approximator of $P$ must also be an absolute approximator of marginals of $P$ w.r.t. $\epsilon$.

**Theorem B.1** (Bounded $D_f$ implies absolute approx. of marginals). *Given two distributions $P(\mathbf{X})$ and $Q(\mathbf{X})$ over a set of variables $\mathbf{X}$ and $0 \le \epsilon \le 1$, if $D_f(P\|Q) < k\epsilon^2$ then for all assignments $\mathbf{y}$ to a subset $\mathbf{Y} \subseteq \mathbf{X}$, $|P(\mathbf{y}) - Q(\mathbf{y})| < \epsilon$.*

*Proof.* Note that while the absolute error of marginals is symmetric between $P$ and $Q$, $f$-divergence between $P$ and $Q$, such as the KL-divergence, is not symmetric. Therefore, we utilize the implications derived in [28], that $D_f(P\|Q) < k\epsilon^2$ then $D_{\mathsf{TV}}(P\|Q) < \epsilon$. Moreover, given that the total variation distance is an $f$-divergence, we know that by the monotonicity property $D_f(P(\mathbf{Y}), Q(\mathbf{Y})) \le D_f(P(\mathbf{X}), Q(\mathbf{X}))$ [30] as $\mathbf{Y} \subseteq \mathbf{X}$. By definition, $\max_{S \subseteq \{0,1\}^n} |P(S) - Q(S)| < \epsilon$. Since these are discrete probability distributions, the event $S = \mathbf{y}$ is valid. Therefore, $\forall \mathbf{y} : |P(\mathbf{y}) - Q(\mathbf{y})| < \epsilon$. $\square$

From the above proof, we immediately derive the following corollary.

**Corollary B.2** (Bounded $D_{\mathsf{TV}}$ implies absolute approx. of marginals). *Given two distributions $P(\mathbf{X})$ and $Q(\mathbf{X})$ over a set of variables $\mathbf{X}$ and $0 \le \epsilon \le 1$, if $D_{\mathsf{TV}}(P\|Q) < \epsilon$, then for all assignments $\mathbf{y}$ to $\mathbf{Y} \subseteq \mathbf{X}$, $|P(\mathbf{y}) - Q(\mathbf{y})| < \epsilon$.*

Since marginals are tractable for decomposable PCs, approximating a target distribution with bounded $f$-divergence using a decomposable PC implies that marginals can be approximated in polynomial-time with bounded absolute error. Previously, Dagum and Luby [12] considered this query for Bayesian networks and showed that there exists a randomized polynomial-time algorithm for the absolute approximation of marginals. Due to this, it is unsurprising that this form of approximation is guaranteed by the more restrictive condition of bounded f-divergence.

Moving to more difficult queries, we next study approximate inference implications for MAP. Let $P(\mathbf{X})$ be a probability distribution over a set of variables $\mathbf{X}$. We say another distribution $Q(\mathbf{X})$ is an absolute approximator of the *maximum-a-posteriori* of $P$ w.r.t. $0 \le \epsilon \le 1$ if: for every assignment $\mathbf{e}$ (called the evidence) to a subset $\mathbf{E} \subseteq \mathbf{X}$, $|\max_{\mathbf{y}} P(\mathbf{y}, \mathbf{e}) - \max_{\mathbf{y}} Q(\mathbf{y}, \mathbf{e})| \le \epsilon$ where $\mathbf{Y} = \mathbf{X} \setminus \mathbf{E}$. We next show that $k\epsilon^2$-$D_f$-approximators are also absolute approximators of MAP w.r.t. $\epsilon$.

**Theorem B.3** (Bounded $D_f$ implies absolute approx. of MAP). *Given two distributions $P(\mathbf{X})$ and $Q(\mathbf{X})$ over a set of variables $\mathbf{X}$ and $0 \le \epsilon \le 1$, if $D_f(P\|Q) < k\epsilon^2$ then for every assignment $\mathbf{e}$ to a subset $\mathbf{E} \subseteq \mathbf{X}$, $|\max_{\mathbf{y} \in \mathbf{Y}} P(\mathbf{y}, \mathbf{e}) - \max_{\mathbf{y} \in \mathbf{Y}} Q(\mathbf{y}, \mathbf{e})| < \epsilon$ where $\mathbf{Y} = \mathbf{X} \setminus \mathbf{E}$.*

---

[1]Sometimes also called the most probable explanation (MPE).

[2]For PCs over Boolean variables, a weaker form of decomposability called *consistency* [31, 6] actually suffices instead of decomposability for both tractable marginal and MAP inference. In this paper, we still focus on classes of PCs that are decomposable as they are the most commonly considered, both as learning targets as well as for characterizing expressive efficiency.

*Proof.* Analogous to Theorem B.1, we utilize the fact that if we have $D_f(P\|Q) < k\epsilon^2$, we know $D_{\text{TV}}(P\|Q) < \epsilon$. Using $D_{\text{TV}}(P\|Q) < \epsilon$, we have $\max_{\mathbf{x}} |P(\mathbf{x}) - Q(\mathbf{x})| < \epsilon$ by definition. Then for all $\mathbf{x}$, $Q(\mathbf{x}) - \epsilon < P(\mathbf{x}) < Q(\mathbf{x}) + \epsilon$. W.l.o.g., suppose $\max P(\mathbf{x}) > \max Q(\mathbf{x})$. Then $|\max P(\mathbf{x}) - \max Q(\mathbf{x})| < (\max Q(\mathbf{x}) + \epsilon) - \max Q(\mathbf{x}) = \epsilon$. Thus, as this was for all $\mathbf{X}$, we can extend this to subsets of $\mathbf{X}$. Finding that for every assignment $\mathbf{e}$ to a subset $\mathbf{E} \subseteq \mathbf{X}$, $|\max_{\mathbf{y} \in \mathbf{Y}} P(\mathbf{y}, \mathbf{e}) - \max_{\mathbf{y} \in \mathbf{Y}} Q(\mathbf{y}, \mathbf{e})| < \epsilon$ where $\mathbf{Y} = \mathbf{X} \setminus \mathbf{E}$. Thus, approximating with bounded $f$-divergence by a deterministic PC implies polynomial-time approximation of MAP with bounded absolute error. $\square$

Again, we restrict this to the special case of total variation distance.

**Corollary B.4** (Bounded $D_{\text{TV}}$ implies absolute approx. of MAP). *Given two distributions $P(\mathbf{X})$ and $Q(\mathbf{X})$ over a set of variables $\mathbf{X}$ and $0 \leq \epsilon \leq 1$, if $D_{\text{TV}}(P\|Q) < \epsilon$ then for every assignment $\mathbf{e}$ to a subset $\mathbf{E} \subseteq \mathbf{X}$, $|\max_{\mathbf{y} \in \mathbf{Y}} P(\mathbf{y}, \mathbf{e}) - \max_{\mathbf{y} \in \mathbf{Y}} Q(\mathbf{y}, \mathbf{e})| < \epsilon$ where $\mathbf{Y} = \mathbf{X} \setminus \mathbf{E}$.*

Thus, a deterministic PC that is an $\epsilon$-$D_{\text{TV}}$-approximator of a distribution $P$ would imply that exact MAP inference w.r.t. this PC grants us tractable approximate MAP inference w.r.t. the original model. However, the converse does not hold: a PC that can be used for approximate MAP inference is not necessarily a good approximation of the full distribution. In particular, we can obtain PCs with identical MAP solutions, but result in large total variation distance due to disjoint support, as seen in Counterexample 1.

Lastly, not all tractable queries for PCs are guaranteed to admit absolute approximation even under the restrictive framework of approximate modeling with bounded distance.

**Theorem B.5** (Bounded $D_{\text{TV}}$ does not imply absolute approx. of conditionals/conditional MAP). *There exists a family of distributions $P$ that have $\epsilon$-$D_{\text{TV}}$- approximators, yet the absolute approximation for conditional marginals and conditional MAP can be arbitrarily large.*

*Proof sketch.* Let $P$ be a probability distribution over $\mathbf{X}$, and let some assignment $\mathbf{e}$ to $\mathbf{E} \subseteq \mathbf{X}$ be evidence. We construct a distribution $Q$ using the mass function $Q(\mathbf{y}^*, \mathbf{e}) = P(\mathbf{e})P(\mathbf{y}^*|\mathbf{e}) + k\epsilon P(\mathbf{e})$ where $\mathbf{y}^*$ maximizes $P(\mathbf{y}|\mathbf{e})$. Then we remove the mass we added to ensure $Q$ is still a valid probability distribution. This allows us to construct a distribution with $D_{\text{TV}}(P\|Q) < \epsilon$ but conditional MAP equals $k\epsilon$ where $k$ can be arbitrarily large. We refer to Appendix E.6 for the full distribution construction and detailed proof. $\square$

Moreover, because relative approximation of conditionals imply their absolute approximation [12], bounded $D_{\text{TV}}$ also does not imply relative approximation of conditionals. However, it is well known that both absolute and relative approximation of conditionals is NP-hard in Bayesian networks [12]. Even though approximate modeling with a bounded $D_{\text{TV}}$ is also an NP-hard task, it still does not guarantee a polynomial-time algorithm for approximating conditional queries. This highlights a key limitation: while tractability of queries is guaranteed by the structural properties of our learned PCs, some queries do not yield "good" approximations for all assignments even after learning within bounded distance.

## C $k$-CONVEX $f$-DIVERGENCES

We rewrite the table from [28] here.

| Divergence | f | $k$ | Domain |
|---|---|---|---|
| relative entropy (KL) | $t \log t$ | $\frac{1}{M}$ | $(0, M]$ |
| total variation | $\frac{|t-1|}{2}$ | $0$ | $(0, \infty)$ |
| Pearson's $\chi^2$ | $(t-1)^2$ | $2$ | $(0, \infty)$ |
| squared Hellinger | $2(1 - \sqrt{t})$ | $M^{-\frac{3}{2}}/2$ | $(0, M]$ |
| reverse relative entropy | $-\log t$ | $\frac{1}{M^2}$ | $(0, M]$ |
| Vincze-Le Cam | $\frac{(t-1)^2}{t+1}$ | $\frac{8}{(M+1)^3}$ | $(0, M]$ |
| Jensen–Shannon | $(t+1) \log \frac{2}{t+1} + t \log t$ | $\frac{1}{M(M+1)}$ | $(0, M]$ |
| Neyman's $\chi^2$ | $\frac{1}{t} - 1$ | $\frac{2}{M^3}$ | $(0, M]$ |
| Sason's $s$ | $\log(s+t)^{(s+t)^2} - \log(s+1)^{(s+1)^2}$ | $2\log(s+M) + 3$ | $[M, \infty), s > e^{-3/2}$ |
| $\alpha$-divergence | $\frac{4\left(1 - t^{\frac{1+\alpha}{2}}\right)}{1 - \alpha^2}, \quad \alpha \neq \pm 1$ | $M^{\frac{\alpha-3}{2}}$ | $\begin{cases} [M, \infty), & \alpha > 3 \\ (0, M], & \alpha < 3 \end{cases}$ |

Table 1: Examples of Strongly Convex Divergences.

## D RECTANGLE PARTITIONS

Rectangle partitions are a powerful tool used in communication complexity to analyze the size of communication protocols. The main idea is to represent the communication protocol for a function $f : \{0,1\}^n \to \{0,1\}$ into a $2^n \times 2^n$ matrix $M_f$ where $M_f[\mathbf{x}, \mathbf{y}] = f(\mathbf{x}, \mathbf{y})$, then partition $M_f$ into a set of monochromatic rectangles which cover the input space of all possible pairs. Here, monochromatic means that a given rectangle covers only the outputs equal to 0 or 1, but not both. This allows us to derive lower bounds on the communication complexity of a function $f$. Furthermore, Bova et al. [1] showed the relation between rectangle covers and partitions to the size of DNNF and d-DNNF formulas. For all definitions below, assume that $\mathbf{X}$ is a finite set of variables.

We begin by describing *partitions* of $\mathbf{X}$, corresponding to our partition of $M_f$.

**Definition D.1** (Partition [1]). A partition of $\mathbf{X}$ is a sequence of pairwise disjoint subsets $\mathbf{X}_i$ of $\mathbf{X}$ such that $\bigcup_i \mathbf{X}_i = \mathbf{X}$. A partition of two blocks $(\mathbf{X}_1, \mathbf{X}_2)$ is *balanced* if $|\mathbf{X}| / 3 \leq \min(|\mathbf{X}_1|, |\mathbf{X}_2|)$.

We can now define *rectangles*:

**Definition D.2** (Combinatorial Rectangle [1]). A rectangle over $\mathbf{X}$ is a function $r : \{0,1\}^{|\mathbf{X}|} \to \{0,1\}$ such that there exists and underlying partition of $\mathbf{X}$, called $(\mathbf{X}_1, \mathbf{X}_2)$ and functions $r_i : \{0,1\}^{|\mathbf{X}|} \to \{0,1\}$ for $i = 1, 2$ such that $r^{-1}(1) = r_1^{-1}(1) \times r_2^{-1}(1)$. A rectangle is *balanced* if the underlying partition is balanced.

Combining many of these rectangles together allows us to cover $M_f$, effectively covering the function $f$. The size of these covers provides lower bounds on the communication complexity of $f$.

**Definition D.3** (Rectangular Cover [1]). Let $f : \{0,1\}^{|\mathbf{X}|} \to \{0,1\}$ be a function. A finite set of rectangles $\{r_i\}$ over $\mathbf{X}$ is called a *rectangle cover* if

$$f^{-1}(1) = \bigcup_i r_i^{-1}(1).$$

The rectangle cover is referred to as a *rectangle partition* if the above union is disjoint. A rectangle cover is *balanced* if each rectangle in the cover is balanced.

To understand how these rectangle partitions relate to d-DNNFs and DNNFs, we utilize the following notions of *certificates* and *elimination*.

**Definition D.4** (Certificate [1])**.** Let $\mathcal{C}$ be a DNNF on $\mathbf{X}$. A *certificate of* $\mathcal{C}$ is a DNNF $T$ on $\mathbf{X}$ such that: $T$ contains the output gate of $\mathcal{C}$; if $T$ contains an $\wedge$-gate, $v$, of $\mathcal{C}$ then $T$ also contains every gate of $\mathcal{C}$ having an output wire to $v$; if $T$ contains an $\vee$-gate of $\mathcal{C}$, then $T$ also contains exactly one gate of $\mathcal{C}$ having an output wire to $v$. The output gate of $T$ coincides with the output gate of $\mathcal{C}$, and the gates of $T$ inherit their labels and wires from $\mathcal{C}$. We let $cert(\mathcal{C})$ denote the certificates of $\mathcal{C}$.

See from the above definition that

$$\mathcal{C}^{-1}(1) = \bigcup_{T \in cert(\mathcal{C})} T^{-1}(1).$$

This is useful tool in relation to rectangle partitions due to the fact that given a DNNF $\mathcal{C}$, $T \in cert(\mathcal{C})$ and gate $g$, then $\mathcal{C}_g^{-1}(1) = \bigcup_{T \in cert(\mathcal{C}_g)} T^{-1}(1)$ where $\mathcal{C}_g$ represents the sub-circuit $\mathcal{C}$ rooted at gate $g$. Then we can represent $\mathcal{C}_g^{-1}(1)$ as a rectangle which separates the variables in the sub-circuit $\mathcal{C}$ rooted at $g$. Using this in conjunction with the *elimination* operation gives us the ability to compute the size of our circuit using rectangles.

**Definition D.5** (Elimination [1])**.** Let $\mathcal{C}$ be a DNNF and $g$ be a non-input gate. Then,

$$\mathcal{C}_{\neg g}^{-1}(1) = \bigcup_{T \in cert(\mathcal{C}) \setminus cert(\mathcal{C}_g)} T^{-1}(1)$$

In the case of a d-DNNF, by determinism we can write $\mathcal{C}_{\neg g}^{-1}(1) = \mathcal{C}^{-1}(1) \setminus \mathcal{C}_g^{-1}(1)$.

Here we provide a short description on the relationship between the size of rectangle covers and Boolean circuits; for the full detailed proofs see [1]. Effectively, start with a d-DNNF $\mathcal{C}$ over variables $\mathbf{X}$ which computes a function $f$. Then, construct $\mathcal{C}^{i+1} = \mathcal{C}_{\neg g_i}^i$ by eliminating $g_i \in \mathcal{C}^i$ until we hit $l \leq |\mathcal{C}|$ such that $\mathcal{C}^l \equiv 0$. It can be shown that $R_i = (\mathcal{C}_{g_i}^i)^{-1}(1)$ is a balanced rectangle over $\mathbf{X}$. The set $\{R_i | i = 0, \ldots, l-1\}$ is then a balanced rectangle partition of $\mathcal{C}$ since $(\mathcal{C}_{\neg g_i}^{i+1})^{-1}(1) = \emptyset$.

Therefore, we can represent the size of d-DNNFs representing functions as the size of a balanced rectangle partition over said function. This implies that an exponential size rectangle partition implies exponentially large d-DNNF.

# E   COMPLETE PROOFS

## E.1   DISTRIBUTION CONSTRUCTION FOR PROPOSITION 3.2

*Proof.* Let $A$ be an event such that $P(A) = \delta$ and for some $K > 0$,

$$Q(\mathbf{x}) = \frac{P(\mathbf{x})}{K}, \ \forall \mathbf{x} \in A.$$

Then, let $Q(\mathbf{x}) = \lambda P(\mathbf{x}), \ \forall \mathbf{x} \in A^c$ for some constant $\lambda$. To ensure that $Q$ is normalized, see that we must have $\sum_{\mathbf{x}} Q(\mathbf{x}) = \sum_{\mathbf{x} \in A} Q(\mathbf{x}) + \sum_{\mathbf{x} \in A^c} Q(\mathbf{x}) = 1$. Therefore, we must have:

$$1 = \sum_{\mathbf{x} \in A} Q(\mathbf{x}) + \sum_{\mathbf{x} \in A^c} Q(\mathbf{x}) = \sum_{\mathbf{x} \in A} \frac{P(\mathbf{x})}{K} + \sum_{\mathbf{x} \in A^c} \lambda P(\mathbf{x}) = \frac{\delta}{K} + \lambda(1 - \sum_{\mathbf{x} \in A} P(\mathbf{x}))$$

$$= \frac{\delta}{K} + \lambda(1 - \delta)$$

Hence $\lambda = \frac{1 - \delta/K}{1 - \delta}$ and thus we can define

$$Q(\mathbf{x}) = \begin{cases} \frac{P(\mathbf{x})}{K}, & \mathbf{x} \in A \\ \frac{1 - \delta/K}{1 - \delta} P(\mathbf{x}), & \mathbf{x} \in A^c \end{cases}$$

Therefore, we just need to check then that the $f$-divergence between $P$ and $Q$ must be bounded.

$$\sum_{\mathbf{x}} Q(\mathbf{x}) f\left(\frac{P(\mathbf{x})}{Q(\mathbf{x})}\right) = \sum_A Q(\mathbf{x}) f\left(\frac{P(\mathbf{x})}{Q(\mathbf{x})}\right) + \sum_{A^c} Q(\mathbf{x}) f\left(\frac{P(\mathbf{x})}{Q(\mathbf{x})}\right)$$

$$= \sum_A \frac{P(\mathbf{x})}{K} f\left(\frac{P(\mathbf{x})}{P(\mathbf{x})/K}\right) + \sum_{A^c} \frac{1-\delta/K}{1-\delta} P(\mathbf{x}) f\left(\frac{1-\delta}{1-\delta/K}\right)$$

$$= \frac{f(K)}{K} \sum_A P(\mathbf{x}) + \frac{1-\delta/K}{1-\delta} f\left(\frac{1-\delta}{1-\delta/K}\right) \sum_{A^c} P(\mathbf{x})$$

$$= \frac{\delta f(K)}{K} + \frac{1-\delta/K}{1-\delta} f\left(\frac{1-\delta}{1-\delta/K}\right) (1-\delta)$$

$$= \frac{\delta f(K)}{K} + \left(1 - \frac{\delta}{K}\right) f\left(\frac{1-\delta}{1-\delta/K}\right)$$

See that as $\delta \to 0$, the above approaches $0 + f(1)$. By definition of $f$-divergence, $f(1) = 0$. Thus, the $f$-divergence—including the total variation distance—between $P$ and $Q$ can be very small, approaching 0, while the relative approximation error stays at a constant factor $K$. $\qquad\square$

## E.2 PROOF OF THEOREM 3.3

*Proof.* We will prove the above by reducing from SAT. Let $f$ be a Boolean formula over $\mathbf{X} = \{X_1, \ldots, X_n\}$ and $\epsilon < \frac{1}{4}$. Define a new Boolean formula $f'$ over $\mathbf{X}$ and $Y$: $f' = (Y \wedge f) \vee (\neg Y \wedge X_1 \wedge \cdots \wedge X_n)$. Clearly $f'$ has $\mathrm{MC}(f) + 1$ models. Let us now define a uniform distribution $P$ over these models of $f'$. Suppose that we can efficiently obtain a probability distribution $Q$ such that $D_f(P\|Q) < k\epsilon^2$, which in turn implies that $D_{\mathsf{TV}}(P\|Q) < \epsilon$. From the definition of total variation distance, $|P(Y=1) - Q(Y=1)| < \epsilon < \frac{1}{4}$.

By construction, if $f$ is unsatisfiable, there is no satisfying model of $f'$ such that $Y = 1$, and thus $P(Y=1) = 0$ and $Q(Y=1) < \frac{1}{4}$. Otherwise, if $f$ is satisfiable, then there are $\mathrm{MC}(f)$ many satisfying model of $f'$ setting $Y = 1$, and thus we have $\mathrm{MC}(f) \geq 1$, $P(Y=1) = \frac{\mathrm{MC}(f)}{1+\mathrm{MC}(f)}$, and

$$Q(Y=1) > \frac{\mathrm{MC}(f)}{1+\mathrm{MC}(f)} - \frac{1}{4} \geq \frac{1}{2} - \frac{1}{4} \geq \frac{1}{4}.$$

Therefore, $f$ is satisfiable if and only if $Q(Y=1) \geq \frac{1}{4}$. In other words, we can decide SAT if we can efficiently compute an $k\epsilon^2$-$D_f$-approximation as a model that supports tractable marginals. $\qquad\square$

## E.3 PRUNING DETERMINISTIC PCS FOR PROOF OF PROPOSITION 4.3

Suppose that $Q$ is a deterministic, decomposable and smooth probabilistic circuit. Given $Q$, wish to prune its edges such that in the resulting (unnormalized) PC $Q'$, $\mathbf{x}$ is in the support of $Q'$ if and only if $Q(\mathbf{x}) < \frac{1}{2^{n+1}}$. We describe our pruning algorithm below.

First, we collect the an upper bound on each edge $(n, c)$ that is the largest probability obtainable by any assignment $\mathbf{x}$ that uses that edge (propagates non-zero value through the edge in the forward pass for $Q(\mathbf{x})$). We denote this $EB(n, c)$, which stands for the Edge-Bound. This can be done in linear time in the size of the circuit using the Edge-Bounds algorithm [8]. This allows us to safely prune any edge whose Edge-Bound falls below a given threshold; i.e., prune edge $(n, c)$ if $EB(n, c) < \frac{1}{2^{n+1}}$.

Note that pruning some edges may cause the edge bounds for remaining edges to be tightened. Thus, we will repeat this process until all $Q(\mathbf{x}) < \frac{1}{2^{n+1}}$ are pruned away. Upon completion of this process, we return back the new pruned circuit $Q'$.

We know that this algorithm halts as there can only be a finite number of $\mathbf{x}$ such that $Q(\mathbf{x}) < \frac{1}{2^{n+1}}$. Moreover, given that we have only deleted edges from a $Q$, our circuit $Q'$ is still a deterministic, decomposable and smooth probabilistic circuit and has size polynomial in the size of $Q$. We are also assured by determinism that if we prune a path $Q(\mathbf{x})$, there exists no other path that can evaluate $Q(\mathbf{x})$ [4]; thus all $Q(\mathbf{x}) < \frac{1}{2^{n+1}}$ are deleted. Furthermore, by the property that there is only one accepting path per assignment $\mathbf{x}$, we know that we do not unintentionally delete any $Q(\mathbf{x}) \geq \frac{1}{2^{n+1}}$.

### E.4 PROOF OF PROPOSITION 4.3

*Proof.* Suppose that $P$ is a uniform distribution over the support given by a Boolean function $f$, and $Q$ a probability distributions over the support given by a Boolean function $h$. Further suppose that $D_{\mathsf{TV}}(P\|Q) < \epsilon$ and $0 \le \epsilon < \frac{1}{8}$. Then, there exists another polynomial sized decomposable and deterministic (unnormalized) PC $Q'$ by pruning the edges of $Q$ such that all assignments $Q(\mathbf{x}) < \frac{1}{2^{n+1}}$ are eliminated from the resulting support of $Q'$. This is obtained using the algorithm from Appendix E.3. We call the new support $g$. We utilize the bound on TV distance between distributions to obtain a bound on weak approximation between their supports as follows.

$$D_{\mathsf{TV}}(P\|Q) = \frac{1}{2}\sum_{x \models f \wedge h}|P(\mathbf{x}) - Q(\mathbf{x})| + \sum_{\mathbf{x} \models f \wedge \neg h}P(\mathbf{x}) + \sum_{\mathbf{x} \models \neg f \wedge h}Q(\mathbf{x}) < \epsilon$$

$$\implies \sum_{x \models f \wedge h}|P(\mathbf{x}) - Q(\mathbf{x})| + \sum_{\mathbf{x} \models f \wedge \neg h}P(\mathbf{x}) + \sum_{\mathbf{x} \models \neg f \wedge h}Q(\mathbf{x}) < 2\epsilon$$

We partition $h$ into the disjoint sets $g$ and $\neg g \wedge h$ (note that every model of $g$ is already a model of $h$).

$$\sum_{\mathbf{x} \models f \wedge g}\left|\frac{1}{\mathrm{MC}(f)} - Q(\mathbf{x})\right| + \sum_{\mathbf{x} \models f \wedge (\neg g \wedge h)}\left|\frac{1}{\mathrm{MC}(f)} - Q(\mathbf{x})\right| + \frac{\mathrm{MC}(f \wedge \neg h)}{\mathrm{MC}(f)} + \sum_{\mathbf{x} \models \neg f \wedge h}Q(\mathbf{x}) < 2\epsilon$$

The LHS of above inequality is again lower bounded by:

$$\sum_{\mathbf{x} \models f \wedge (\neg g \wedge h)}\left|\frac{1}{\mathrm{MC}(f)} - Q(\mathbf{x})\right| + \frac{\mathrm{MC}(f \wedge \neg h)}{\mathrm{MC}(f)} + \sum_{\mathbf{x} \models \neg f \wedge h}Q(\mathbf{x})$$

$$> \sum_{\mathbf{x} \models f \wedge (\neg g \wedge h)}\left|\frac{1}{2^n} - \frac{1}{2^{n+1}}\right| + \frac{\mathrm{MC}(f \wedge \neg h)}{\mathrm{MC}(f)} + \sum_{\mathbf{x} \models \neg f \wedge g}Q(\mathbf{x})$$

$$> \sum_{\mathbf{x} \models f \wedge (\neg g \wedge h)}\left|\frac{1}{2^n} - \frac{1}{2^{n+1}}\right| + \frac{\mathrm{MC}(f \wedge \neg h)}{\mathrm{MC}(f)} + \frac{\mathrm{MC}(\neg f \wedge g)}{2^{n+1}}$$

as $\mathrm{MC}(f) \le 2^n$ and $Q(\mathbf{x}) < 1/2^{n+1}$ for every $\mathbf{x} \models \neg g \wedge h$. Thus,

$$2\epsilon > \sum_{\mathbf{x} \models f \wedge (\neg g \wedge h)}\left|\frac{1}{2^n} - \frac{1}{2^{n+1}}\right| + \frac{\mathrm{MC}(f \wedge \neg h)}{\mathrm{MC}(f)} + \frac{\mathrm{MC}(\neg f \wedge g)}{2^{n+1}}$$

$$> \frac{\mathrm{MC}(f \wedge (\neg g \wedge h))}{2^{n+1}} + \frac{\mathrm{MC}(f \wedge \neg h)}{\mathrm{MC}(f)} + \frac{\mathrm{MC}(\neg f \wedge g)}{2^{n+1}}$$

$$> \frac{\mathrm{MC}(f \wedge (\neg g \wedge h))}{2^{n+1}} + \frac{\mathrm{MC}(f \wedge \neg h)}{2^{n+1}} + \frac{\mathrm{MC}(\neg f \wedge g)}{2^{n+1}} = \frac{\mathrm{MC}(f \wedge \neg g) + \mathrm{MC}(\neg f \wedge g)}{2^{n+1}},$$

implying $\mathrm{MC}(f \wedge \neg g) + \mathrm{MC}(\neg f \wedge g) < 2^{n+2}\epsilon = 2^n(4\epsilon)$. Note that the last equality above is due to $\neg h$ implying $\neg g$. Thus, $g$ is a $4\epsilon$-weak-approximator of $f$, and we can represent it as a polynomially sized d-DNNF by taking the decomposable and deterministic PC $Q'$ and converting it to a Boolean circuit ($\oplus$ and $\otimes$ to $\vee$ and $\wedge$ respectively) [4, 41]. $\square$

### E.5 PROOF OF THEOREM 4.1

*Proof of Theorem 4.1.* Suppose we have a deterministic, decomposable PC, $Q$ that is an $\epsilon$-$D_{\mathsf{TV}}$-Approximator of $P_n$ where $\epsilon = (\frac{1}{16} - \Omega(1/Poly(n^2)))$ and $\eta = \frac{1}{(1-1/\sqrt{2})2^{n^2}}$. Consider the uniform distribution $U$ over $S_n$. By the triangle inequality, $D_{\mathsf{TV}}(U\|Q) \le D_{\mathsf{TV}}(U\|P_n) + D_{\mathsf{TV}}(P_n\|Q) < D_{\mathsf{TV}}(U\|P_n) + \epsilon - \Omega(1/Poly(n^2))$. By [1, Proposition 7], the DNNF constructed to represent the Sauerhoff function is a disjunction of two OBDDs, which respectively represent $RT_n, CT_n$. As each OBDD is a deterministic circuit, their PC counterpart (OR to sum and AND to product) will still represent the same Boolean functions [4]. Thus, the non-deterministic PC representing $P_n$ based on this construction only has one non-deterministic sum node at the root, and can return values at most 2. This allows us to see that $D_{\mathsf{TV}}(U\|P) \le \left|\frac{1}{|S_n|} - \frac{2}{|S_n|+1}\right| < \frac{1}{|S_n|} < \frac{1}{(1-\beta)2^{n^2}} < \eta$ because $|S_n| > (1-\beta)2^{n^2}$ for $\beta < 1/\sqrt{2}$. $|S_n| > (1-\beta)2^{n^2}$ is derived from the low 0-density property of $S_n$ under the uniform distribution [35]. Thus, $D_{\mathsf{TV}}(U\|Q) < \epsilon - \Omega(1/Poly(n^2)) + \eta$. Since, $\Omega(1/Poly(n^2))$ is unspecified, we can add $\eta$ to this and more simply say $D_{\mathsf{TV}}(U\|Q) < \epsilon - \Omega(1/Poly(n^2))$. By Proposition 4.3, we can construct a d-DNNF $\mathcal{C}'$ from $Q$—by replacing $\oplus$ and $\otimes$ to $\vee$ and $\wedge$ respectively—that is a $(\frac{1}{4} - \Omega(1/Poly(n^2)))$-weak-approximation of $S_n$. Thus, by Proposition 4.4, $|Q'| = 2^{\Omega(n)}$. $\square$

### E.6 PROOF OF THEOREM B.5

*Proof.* Let $P$ be some distribution over $\mathbf{X}$ and $\mathbf{Y}, \mathbf{E} \subset \mathbf{X}$ such that $\mathbf{Y} = \mathbf{X} \setminus \mathbf{E}$. We know that for any assignment $\mathbf{y}, \mathbf{e}$ such that $P(\mathbf{e}) > 0$, we have $P(\mathbf{y}, \mathbf{e}) = P(\mathbf{e})P(\mathbf{y}|\mathbf{e})$. Let $k$ such that for some $\mathbf{e}$, we have $P(\mathbf{e}) < 1/k$. We construct another distribution $Q$ such that $Q(\mathbf{y}^*, \mathbf{e}) = P(\mathbf{e})P(\mathbf{y}^*|\mathbf{e}) + k\epsilon P(\mathbf{e})$ where $\mathbf{y}^*$ maximizes $P(\mathbf{y}|\mathbf{e})$. Also for another assignment $\mathbf{y}_1$, define $Q(\mathbf{y}_1, \mathbf{e}) = P(\mathbf{e})P(\mathbf{y}_1|\mathbf{e}) - k\epsilon P(\mathbf{e})$. For all other points we keep $P(\mathbf{y}, \mathbf{e}) = Q(\mathbf{y}, \mathbf{e})$. Note that $Q$ is normalized by construction. Thus, we calculate the total variation distance as

$$\frac{1}{2}\left( |P(\mathbf{e})P(\mathbf{y}^*|\mathbf{e}) - P(\mathbf{e})P(\mathbf{y}^*|\mathbf{e}) - k\epsilon P(\mathbf{e})| + |P(\mathbf{e})P(\mathbf{y}_1|\mathbf{e}) - (P(\mathbf{e})p(\mathbf{y}_1|\mathbf{e}) + k\epsilon P(\mathbf{e}))| \right)$$

$$= \frac{1}{2}\left( |-k\epsilon P(\mathbf{e})| + |k\epsilon P(\mathbf{e}))| \right) = k\epsilon P(\mathbf{e}) < \epsilon.$$

Then, as $P(\mathbf{e})$ shrinks we can allow $k\epsilon$ to grow much larger, so long as $Q(\mathbf{y}^*, \mathbf{e})$ still represents a true probability distribution. This in turn leads to an unbounded conditional MAP:

$$|P(\mathbf{y}^*|\mathbf{e}) - Q(\mathbf{y}^*|\mathbf{e})| = |P(\mathbf{y}^*|\mathbf{e}) - P(\mathbf{y}^*|\mathbf{e}) - k\epsilon| = k\epsilon$$

Furthermore, if $\mathbf{y}^*$ is the same for both $P, Q$ then the implication also extends to conditionals. $\qquad\square$

## F  COUNTEREXAMPLES

**Counterexample 1** (Bounded MAP doesn't imply bounded $D_{\mathsf{TV}}(P\|Q)$). Suppose that $|\max_{\mathbf{x}} P(\mathbf{x}) - \max_{\mathbf{x}} Q(\mathbf{x})| < \epsilon$, where $P$ is a distribution over a set of $n$ variables and $Q$ the same distribution over a set of $n$ variables such that the support of $Q$ is disjoint from the support of $P$. Then, $|\max_{\mathbf{x}} P(\mathbf{x}) - \max_{\mathbf{x}} Q(\mathbf{x})| = 0$, but $D_{\mathsf{TV}}(P\|Q) = 1$ as the supports are disjoint.