# OpenReview forum: "On the Hardness of Approximating Distributions with Probabilistic Circuits"
_auai.org/UAI/2025/Workshop/TPM — TPM 2025_

### Official Review · Reviewer_orNk · 2025-06-10
**Solid Theoretical Work on Approximating Distributions with PCs**

**Rating:** 3

**Review:**

This work provides theoretical insights into approximating probability distributions with PCs, proving that it is NP-hard to approximate a distribution within a bounded f-divergence using any model that has tractable marginal inference. The authors furthermore derive an unconditional, exponential size separation between decomposable PCs and decomposable & deterministic PCs for approximate modeling.

The manuscript is well written and structured, and provides a wide array of additional details and full proofs in the appendix.

---

### Official Review · Reviewer_UX1H · 2025-06-16
**Very nice addition to the growing literature on probabilistic circuits**

**Rating:** 3

**Review:**

Overall this is a very nice contribution to the growing body of work on theoretical properties of tractable probabilistic models, especially those ones expressed as probabilistic circuits.

One idea to slightly strengthen the paper is to mention that the result showing that approximating any probability distribution with a model that allows for tractable marginals is NP-hard, would be to briefly mention explicitly that this also holds for non-monotone models. Such as the ones introduced in [1,2,3].

Other than that the paper would be a very nice addition to the workshop's program.


[1] Loconte, Lorenzo, Stefan Mengel, and Antonio Vergari. "Sum of squares circuits." Proceedings of the AAAI Conference on Artificial Intelligence. Vol. 39. No. 18. 2025.

[2] Wang, Benjie, and Guy Van den Broeck. "On the relationship between monotone and squared probabilistic circuits." Proceedings of the AAAI Conference on Artificial Intelligence. Vol. 39. No. 20. 2025.

[3] Pedro Zuidberg Dos Martires. "A Quantum Information Theoretic Approach to Tractable Probabilistic Models" In Proceedings of the Uncertainty in Artificial Intelligence Conference, 2025.